# New seasonal measurement with stability and clustering seasonal patterns: A case study in Japan from 2011 to 2019

Yu Ogasawara 👤 *

Department of Tourism Science, Tokyo Metropolitan University, Hachioji, Tokyo, Japan

* ogayu@tmu.ac.jp

## Abstract

Seasonality of tourism demand witnesses fluctuations over multiple years. The fluctuations and seasonality often cause seasonal pattern changes. This study presents a new seasonal measurement that considers the stability of seasonal patterns. The measurement is based on a seasonal index and expressed interval-valued data, which are a kind of symbolic data. As a case study, the new measurements are calculated from Japanese overnight data from 2011 to 2019 and the results classify the seasonal patterns of 47 prefectures using a hierarchical clustering method for interval-valued data based on Ward's method. The analysis results indicate that there are differences in not only seasonal patterns but also their stability between domestic overnight demand and inbound overnight demand. The analysis procedure suggested in this study could be helpful in organizing numerous other unstable seasonal patterns.

## Introduction

The seasonality of tourism demand is an important research topic in the academic fields and industries of tourism and hospitality. Seasonality of demand is widely recognized in economic fields [1] and has long been studied in tourism research. The temporal imbalance of tourism demand causes various problems in the tourism and hospitality industries. For example, problems include unstable employment, inefficiency of investment, and exhaustion of local communities. Although recent tourism studies dealing with tourism demand have mainly focused on forecasting [2], research focusing on seasonality of tourism has also been widely conducted for the development of regional economies, tourism marketing, and policy suggestions.

There are various kinds of data to observe tourism demand. These include, for example, visitors to tourism facilities, air passengers, and room occupancy rate. If the number of analysis subjects (e.g., tourist facilities, hotels, tourism destinations, countries, etc.) is too high, then it is difficult to compare their seasonal patterns and organize data because the data for analysis subjects are time-series data. Therefore, for the problem, many measurements of seasonality have been proposed in tourism studies [3–5]. By utilizing a seasonal measurement, time-series data can be summarized as a real number; nevertheless, we need to choose an appropriate

**Data Availability Statement:** All data are available from the Japan Tourism Agency database (https://www.mlit.go.jp/kankocho/siryou/toukei/shukuhakutoukei.html).

**Funding:** This work was supported by JSPS KAKENHI Grant Number JP20K20080. There was no additional external funding received for this study.

**Competing interests:** The authors have declared that no competing interests exist.

measurement from the proposed seasonal measurements in accordance with our research object.

One of the basic analysis approaches for seasonality is to observe the seasonal patterns of analysis subjects. When conducting exploratory analysis as a first step to gaining an overview, it might be natural to check patterns of seasonality of the analysis subjects and compare the patterns, remembering that the number of the analysis subjects is typically greater than one. If the sample size of the seasonality datapoints is moderate, then we can analyze a simple line graph with time-series data expressing seasonal patterns [6–10]. This is called the graphical analysis in this study. The graphical analysis allows for the direct verification of seasonal peaks, troughs, and shoulder seasons. This method is simple, but a powerful tool for intuitively capturing seasonal patterns of analysis subjects. The seasonal index is a useful seasonal measurement and is often used in tourism studies [4, 6, 9, 11]. The seasonal index consists of averages of seasonal factors by month and is derived from a time series decomposition method and denotes the pure seasonality element of time-series data that does not include trend and residual factors.

Although the seasonal index summarizes the seasonal factors into a 12-dimension real number vector, the measurement does not include information on changes in seasonality because the measurement is presented in terms of averages of seasonal factors by month. This implies that using a seasonal index to determine the seasonal pattern means treating the seasonal pattern as a deterministic one. Even though it has been reported that changes in seasonal patterns tend not to be generally considerable [6], seasonality changes commonly occur [9, 12]. In order to check seasonal patterns by considering their stability over the years, a few tourism studies use seasonal plots or cycle plots that express monthly seasonality and the changes in seasonal factors for each month by using a bar graph or line graph [4, 13]. However, if the number of unstable seasonal patterns that we need to analyze increases, it becomes an even more onerous task to compare and organize these seasonal patterns because the number of generated graphs becomes large.

To tackle this problem, this study suggests a new seasonal measurement that is based on a seasonal index and interval-valued data, which are one of the symbolic data [14]. Symbolic data are a complex data type that expresses the uncertainty of data, and the interval-valued data are presented by interval numbers but not real numbers. The interval-valued seasonal measurement includes information on fluctuations in seasonality over multiple years. In addition, this study introduces a hierarchical clustering method based on Ward's method for the measurement proposed by Ogasawara and Kon [15]. A dendrogram generated by the clustering method reveals the whole picture of analysis subjects, which are presented by interval-valued seasonal measurements. From the new measurement and the clustering method, we can intuitively classify seasonal patterns and compare among them by considering fluctuations of the seasonal patterns.

This study also shows interval-valued seasonal indices computed from Japanese overnight data from 2011 to 2019 and clustering results obtained by the clustering method suggested by Ogasawara and Kon [15], as a case study. Japanese overnight data consist of domestic and inbound guests by prefecture in Japan. As a result, changes in seasonal patterns of domestic guests are negligible and uniform for all prefectures, although the seasonal amplitude of domestic guests in each year is larger than that of inbound guests. Furthermore, the diversity of seasonal patterns for inbound guests among prefectures is higher than that for domestic guests. From the results obtained from the cluster analysis, clusters for domestic guests were divided by slight differences in seasonal patterns with strong stability. Classified results for inbound guests are strongly affected by the locations of prefectures, and the difference in seasonal patterns among the clusters is larger than that of domestic guests. Hence, the stable and

uniform patterns of seasonality of domestic guests might be determined by institutional factors. The seasonal patterns of inbound guests might be determined by the existence of major ski resorts and marketing policies of destination marketing/management organizations for the East Asia tourism market.

Fuzzy theory and rough set theory have been applied in tourism studies dealing with forecasting tourism demand to consider uncertainty [16]. Similarly, this study employs interval-valued data, a kind of symbolic data used to express uncertainty and to deal with the stability of seasonal patterns of tourism demand. The Japanese case study showed that seasonal patterns of Japanese inbound guests by prefecture are less stable than those of domestic guests. This phenomenon has also been detected in Italy [17]. If we need to analyze mutable seasonal patterns for many subjects over multiple years, then both the interval-valued seasonal index and hierarchical clustering method introduced in this study can be helpful tools to present a complete picture of the unstable seasonal patterns.

In this study, an interval-valued seasonal index is suggested as a new seasonal measurement and we show its effectiveness through a Japanese case study. The study also reveals characteristics including instability of seasonality of Japanese overnight demand especially relating to unstable inbound demand. The interval-valued seasonal index can capture a cluster that cannot be observed with the traditional seasonal index. The cluster consists of prefectures in the Tohoku area in Japan which was devastated because of a 9.0 magnitude earthquake and a tsunami that followed in March 2011 [18, 19]. In this area, inbound demand rapidly increased because of the first worldwide destination campaign organized by Japan in 2016.

In the next section, seasonality in tourism demand and its measurement suggested in tourism studies are introduced. Further, seasonal patterns expressed by seasonal measurements and their changes are discussed. In the third section, a new seasonal measurement, called the interval-valued seasonal index, is defined and explained. The hierarchical clustering method for interval-valued data suggested by Ogasawara and Kon [15] is briefly introduced in the fourth section. In the fifth section, overnight data in the case study and its overview are shown. Interval-valued seasonal indices by prefecture in Japan and clustering results are also indicated, and the results are discussed.

## Seasonality

A number of tourism studies have dealt with seasonality of demand in tourism industries. It is well known that the intensity fluctuation of seasonal demand leads to an increasing degree of difficulty in managing the suppliers'side [20, 21]. In the accommodation sector with a fixed capacity, the difficulty of management is directly related to revenue [3]. From the viewpoint of revenue management, which is a field of management aimed at increasing revenue in industries with fixed capacity, large fixed products, and perishable products, such as airlines, hotels, and rental cars [22], it is natural that the supplier's side pays attention to seasonality of tourism demand because the effect of booking control on expected revenue is magnified by increasing demand [23]. The difference between the peak and trough of tourism demand is also important for management and investment in tourism industries because it is not generally easy to increase the capacity of facilities or transport infrastructure to handle tourism demand. If a facility cannot fully cover tourism demand in a certain period, then the manager of the facility might consider adding capacities. However, it often leads to an increase in excess capacity during the off-season. An excessive difference in demand between peak season and off season is undesirable for tourism-related facilities or local regions because it causes inefficiency in development or investment in tourism industries [24] and instability in labor markets [2, 25].

## Seasonal measurement

If an analysis subject has an occupancy rate and we can access the data, then we may directly use the value that is standardized between 0 and 1 or 100 [3]. However, if given data also have scale or a trend, tourism studies often use the time-series decomposition method to isolate and quantify seasonal components (see [4, 6, 7, 9, 11] for details). In this method, it is assumed that the time-series data $O_t$ at time $t \in \{1, \cdots, T\}$ consist of trend-cycle $TC_t$, seasonal $S_t$, and residual $R_t$ components. There are two models: an additional model and a multiplicative model, owing to the differences in relationships among the components. In this study, a multiplicative model

$$O_t = TC_t \cdot S_t \cdot R_t, t = 1, \cdots, T \qquad (1)$$

is employed. The seasonal component $S_t$ is called the seasonal factor and is used as an indicator for seasonal measurements. $S_t$ is interpreted as an amplifier for demand, that is, $S_t = 1$, $S_t > 1$, and $S_t < 1$ corresponds to no seasonality, positive seasonal effect, and negative seasonal effect in $t$, respectively.

This study assumes that time-series data are monthly data to simplify notations. Let $S_{i,y}$, $i = 1, \cdots, 12$, $y = 1, \cdots, Y$ denote seasonal factors driven from the time-series data, where $i$ and $y$ are indices for month and year, respectively. Note that the seasonal factor is still a time-series formula. It can be summarized as some measurements focusing on the seasonal pattern or amplitude of demand [4], to easily make graphs and apply it to various data analysis methods. The definitions of seasonal measurements using seasonal factors are, for examples, as following, seasonal index: $\hat{S}_i = \sum_{y=1}^{Y} S_{i,y}/Y$, $i = 1, \cdots, 12$, seasonal peak: $S_y^{max} = \max_{i \in \{1, \cdots, 12\}} S_{i,y}$, $y = 1 \cdots, Y$, seasonal ratio: $S_y^{ratio} = \max_{i \in \{1, \cdots, 12\}} S_{i,y}/\min_{i \in \{1, \cdots, 12\}} S_{i,y}$, $y = 1, \cdots, Y$ and seasonal range: $S_y^{range} = \max_{i \in \{1, \cdots, 12\}} S_{i,y} - \min_{i \in \{1, \cdots, 12\}} S_{i,y}$, $y = 1, \cdots, Y$ [4]. A popular seasonal measurement that does not include the seasonal factor is the Gini coefficient, which has been used in economics as a traditional measurement [24].

## Seasonal pattern

Seasonal patterns used in tourism studies are often identified by seasonal peak and seasonal trough [6, 7]. Furthermore, a simple and effective way to intuitively capture seasonal patterns is to observe the shape of the seasonal index [11, 26], which is intrinsically the same as graphical analysis by line graph with seasonal factors [9]. Koenig and Bischoff [8] use graphical analysis to indicate seasonal patterns although they use indicators provided by principal component analysis and not seasonal factors.

The definitions of seasonal measurements mentioned in this study imply that seasonal patterns shown by seasonal factors can change over time. While the change in the seasonal pattern is not generally significant [6], it is a common occurrence [9, 12]. Sørensen [12] reported a change in seasonality for hotels' overnight demand in Denmark from 1970 to 1996 by unit root tests. As a more short-term case, Cuccia and Rizzo [9] showed a change in seasonality in Sicily from 1998 to 2006 by using the F-test for dynamic seasonality. The major causes of seasonality in tourism demand are climate, weather, social customs, holidays, business customs, calendar effects, and supply side constraints [2]. Considering that these occurrences are subject to global warming, changing social structures, and changing business environments, it is not surprising that seasonal demand patterns change over time. In addition, the increasing use of retrieval services and online services also affects seasonality [27]. If we need to find a seasonal pattern for a region by taking into account changes in the pattern over the years, seasonal plots [13] and cycle plots [4], which show transitions of seasonal factors for each month, have been

suggested as graphical analysis tools. However, analyses using these graphs do not allow easy observation and organization of multiple seasonal patterns at once. If we use seasonal indices to see patterns for a number of analysis subjects, then we can classify them by cluster analysis with the seasonal indices as input values and reveal the derived clusters on a map, which leads to capturing a complete picture of seasonality for the subjects. In fact, Koenig and Bischoff [8] adopted a similar approach. However, we cannot include information on changes in seasonal patterns in the procedure because the information is lost by calculating seasonal indices even if the seasonal patterns are unstable. Therefore, it is not easy to express many seasonal patterns with their changes and mechanically classify the patterns by using seasonal measurements that have been suggested in tourism studies.

## Interval-valued seasonal index

Interval-valued data are a type of symbolic data that are extended from classic data to treat complex conditions [14, 28]. For instance, let us assume that we obtain one's contact address and a city of the address is "Tokyo." Then, the city name corresponds to a classical categorical value "Tokyo." However, if one's residential city is "Tokyo" or "Yokohama" because of the fact that one has two residences and lives in Yokohama only on weekends, then we may need to treat the city data as a set of values: {*Tokyo, Yokohama*}. The set of values is called multi-valued data, which are a type of symbolic data. In another case, when one's weight is between 60 kg and 61.5 kg, we can define that the data is a closed set [60, 61.5]. The closed set is called the interval-valued data, which is defined as

$$X = [\underline{X}, \overline{X}] = \{x \in \mathbb{R} : \underline{X} \leq x \leq \overline{X}\} \tag{2}$$

where $\mathbb{R}$ denotes the set of real numbers.

The new seasonal measurement based on interval-valued data and a seasonal index is presented as follows:

**Definition 1**. *For the given seasonal factors $S_{i,y}$, $i = 1, \cdots, 12$, $y = 1, \cdots, Y$,*

$$I_i = [\underline{S_i}, \overline{S_i}], i = 1, \cdots, 12 \tag{3}$$

*is called an interval-valued seasonal index for a month i, where $\underline{S_i} = \min_{y \in \{1, \cdots, Y\}} S_{i,y}$ and $\overline{S_i} = \max_{y \in \{1, \cdots, Y\}} S_{i,y}$, $i = 1, \cdots, 12$. In addition, $\boldsymbol{I} = (I_1, \cdots, I_{12})$ is called an interval-valued seasonal index.*

From Definition 1, an interval-valued seasonal index for an analysis subject is addressed using 12-dimensional interval-valued data. The interval-valued seasonal index has maximums and minimums of seasonal factors for each month. $I_i$ in Eq (3) indicates the range of observed seasonal factors for a month *i* from year 1 to *Y*. D'Urso and Leski [29] introduce that interval-valued data are used in three situations: (1) data cannot be shown as a traditional real value for example, daily temperatures or blood pressures, (2) we do not have a true single valued data but an interval value including the true value because of a lack of knowledge, and (3) we have only aggregated data whose values are intervals. The interval-valued seasonal index is defined in situation (1), which means that a seasonal pattern shown by the seasonal index over multiple years needs to be expressed by interval-valued data because the pattern can be changed through various motions. Fig 1 shows an image of transforming seasonal factors into an interval-valued seasonal index and a seasonal pattern addressed by an interval-valued seasonal index. In Fig 1, the number of months is only five, and *Y* = 2 for simplification.

Note that the measurement does not provide an increase or decrease in seasonal factors, although the interval-valued seasonal index can present the change of seasonal factors for each

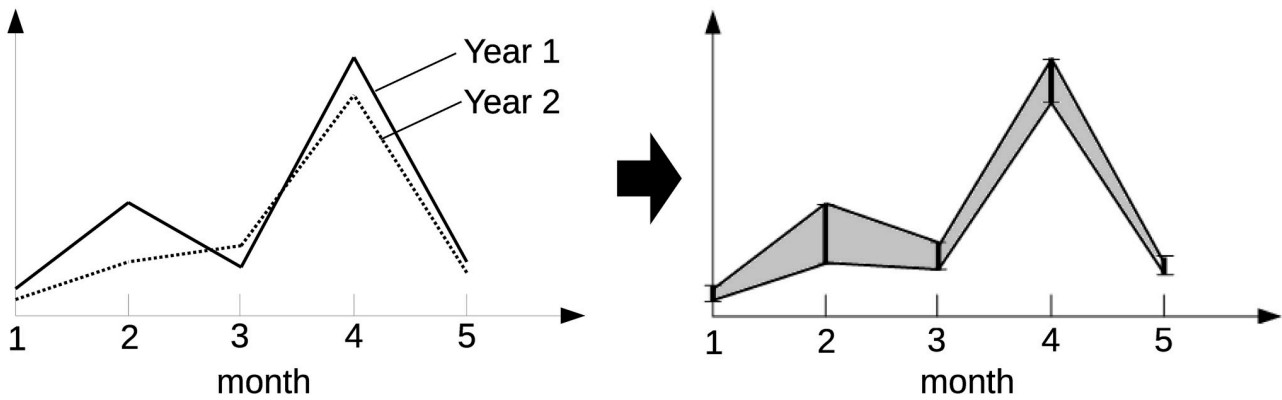

**Fig 1. An image of transforming seasonal factors to an interval-valued seasonal index.**

month as intervals, as seen at months 3 and 4 in Fig 1. Let $w(I_i)$ be the width of the interval-valued seasonal index for a month $i$; that is, $w(I_i) = \overline{I_i} - \underline{I_i}$. In Fig 1, we can find $w(I_2) > w(I_3)$, which can be interpreted as the stability of seasonality in month 3 being smaller than that in month 2 through years 1 and 2.

## Clustering method for interval-valued seasonal indices

When we conduct an exploratory analysis of seasonal patterns expressed by interval-valued seasonal indices without any assumptions, a dendrogram given by a hierarchical clustering method is helpful to look at complete picture of those patterns [30]. Clustering methods for interval-valued data have been studied in the field of pattern recognition [31–34]. Ogasawara and Kon [15] suggested a hierarchical clustering method for interval-valued data based on Ward's method. This study briefly introduces a hierarchical clustering method suggested by Ogasawara and Kon [15] and uses the method for classifying Japanese seasonal patterns as a case study addressed in the next section.

Let $\boldsymbol{I}_u = (I_{1,u}, \cdots, I_{12,u})$, $u \in \{1, \cdots, N\}$ be the interval-valued seasonal index for subject $u$. In the clustering method introduced in this study, the squared Euclidean Hausdorff distance, which is extended from the squared Euclidean distance, is employed instead of the squared Euclidean distance used in Ward's method. The squared Euclidean Hausdorff distance between the interval-valued seasonal indices $\boldsymbol{I}_u$ and $\boldsymbol{I}_v$ is expressed as follows:

$$H^2(\boldsymbol{I}_u, \boldsymbol{I}_v) = \sum_{i=1}^{12} (\max\{|\underline{I_{i,u}} - \underline{I_{i,v}}|, |\overline{I_{i,u}} - \overline{I_{i,v}}|\})^2. \qquad (4)$$

Using the distance function, the sum of the squared deviations of a cluster $G_a$ is

$$ESS(G_a) = \sum_{I \in G_a} H^2(\boldsymbol{I}, E(G_a)) \qquad (5)$$

where $E(G_a) = 1/|G_a| \sum_{\boldsymbol{I} \in G_a} \boldsymbol{I} = 1/|G_a| [\sum_{\boldsymbol{I} \in G_a} \underline{I}, \sum_{\boldsymbol{I} \in G_a} \overline{I}]$, and $|G_a|$ is the number of individuals in $G_a$. $E(G_a)$ is called the interval-valued mean of $G_a$. For two clusters $G_a$ and $G_b$ such that $G_a \cap G_b = \phi$, dissimilarity is defined as

$$\Delta ESS(G_a, G_b) = ESS(G_a \cup G_b) - ESS(E_a) - ESS(E_b). \qquad (6)$$

The form of Eq (6) corresponds to a dissimilarity in Ward's method. We employ the dissimilarity and obtain the result of cluster analysis for interval-valued seasonal indices as a dendrogram by applying the dissimilarity (6) to an algorithm of Ward's method.

## Japanese case study

Japan is ranked as a top-ten tourism destination in the world, as reported by the World Tourism Organization of the United Nations (UNWTO) in 2020 [35]. The number of international tourist arrivals grew at about 3.7 times the rate over 2010 to 2019 [35]. However, there have been few studies on inbound tourism in Japan [36]. Although we can find research that analyzes the influence of climatic and economic variables on seasonality of tourism demand in several major Japanese tourism destinations [37] as a reference, little attention has been paid to the seasonality of tourism demand in Japan.

In this section, actual interval-valued seasonal indices derived from Japanese overnight data of Overnight Travel Statistics Survey over the period January 2011 to December 2019, which were collected by the Japan Tourism Agency are demonstrated. The overnight data consist of two items: domestic guests and inbound guests. In addition, each item has two sectors: 1) the number of overnight stays in hotels where more than half the guests stay for tourism or recreation purposes and 2) a number of overnight stays in hotels where more than half the guests stay for business purposes. Therefore, there are four items that are a combination of domestic or inbound guests and the two purposes. In this study, hotels corresponding to 1) and 2) denote "hotels for tourism" and "hotels for business," respectively. Fig 2 shows a stacked graph for the four items from 2011 to 2020.

It is obvious that seasonality exists on all items of overnights but vanishes in 2020 because of the COVID-19 pandemic. Hence, this study removed overnight data from the analysis subject in 2020. Further, while inbound visitors increased until 2019, as reported by the UNWTO, we can observe that domestic guests still dominate overnight stays in Japan.

## Japanese seasonality

Even though we can clearly find Japanese seasonal patterns from Fig 2, a comparison of patterns of seasonality among those items is unclear. Figs 3 and 4 show the seasonal factors for

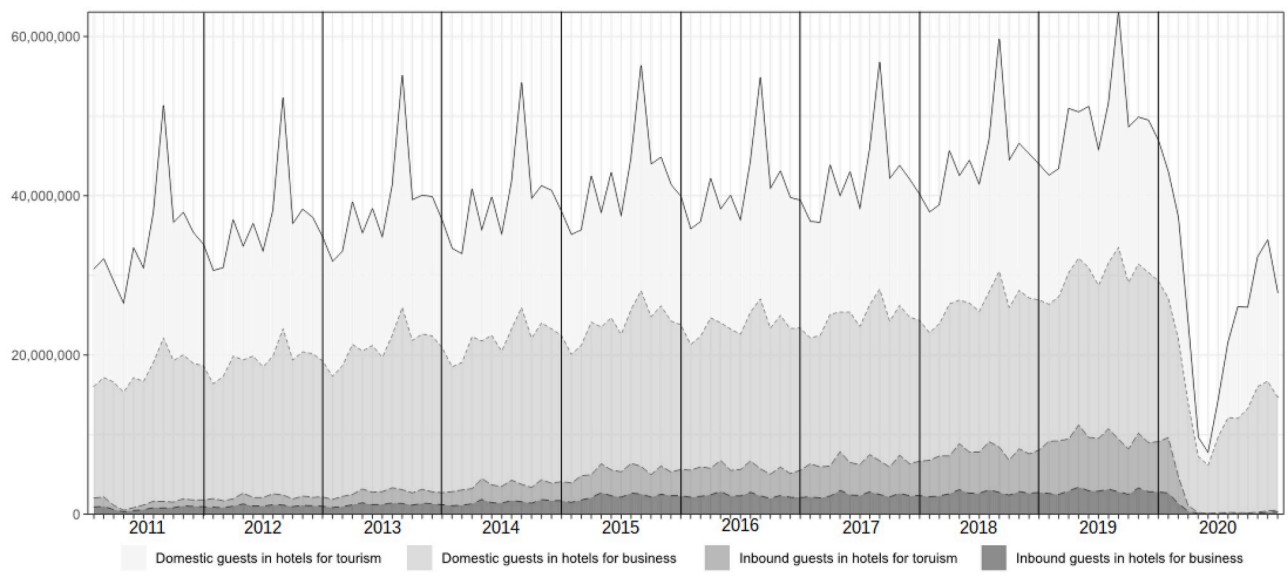

**Fig 2. A stacked graph for Japanese overnights data over 2011 to 2020.**

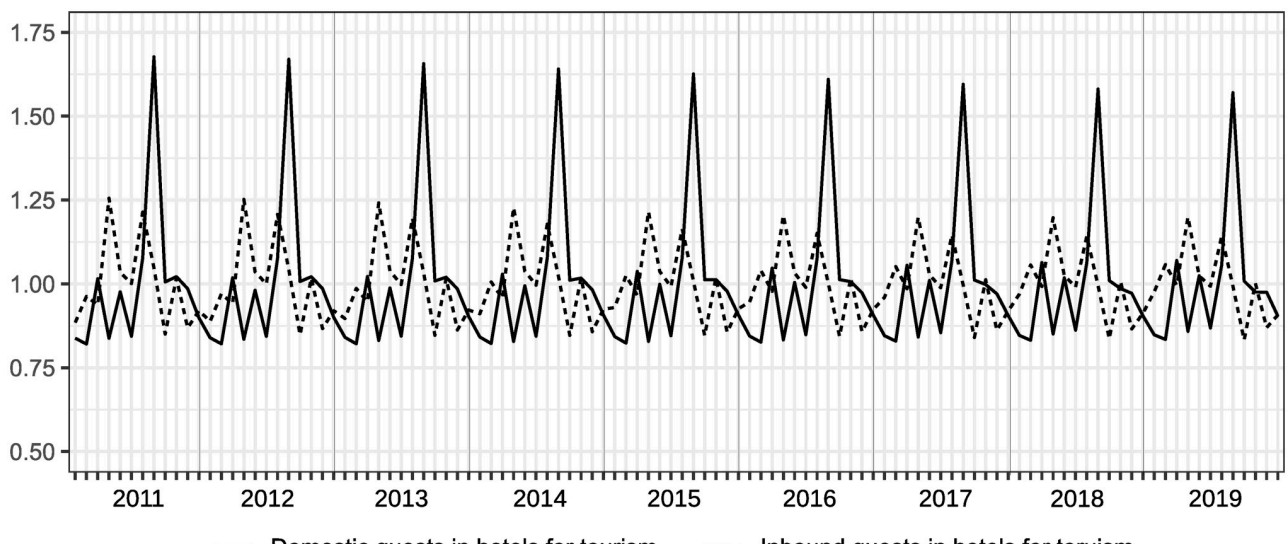

**Fig 3. Seasonal factors of overnight guests in hotels for tourism over 2011 to 2019.**

each item from 2011 to 2019, which are computed by X13-ARIMA-Seat with a multiplicative model and X11 option to obtain the results of statistical seasonal analysis.

Although the scale of the seasonal factors between hotels for tourism and those for business are different among domestic guests, patterns of seasonality are similar to each other. For inbound guests, seasonal patterns between hotels for tourism and those for business are slightly different; specifically, there is a small peak in February in Fig 3, but not in Fig 4, while the scales are almost the same. The main peak was in August, and shoulder peaks were observed in March, May, and October in the case of domestic guests. In terms of inbound guests, peaks were observed in hotels for tourism in February, April, July, and October, and hotels for business in April, July, and October. Note that the seasonal peaks of domestic guests

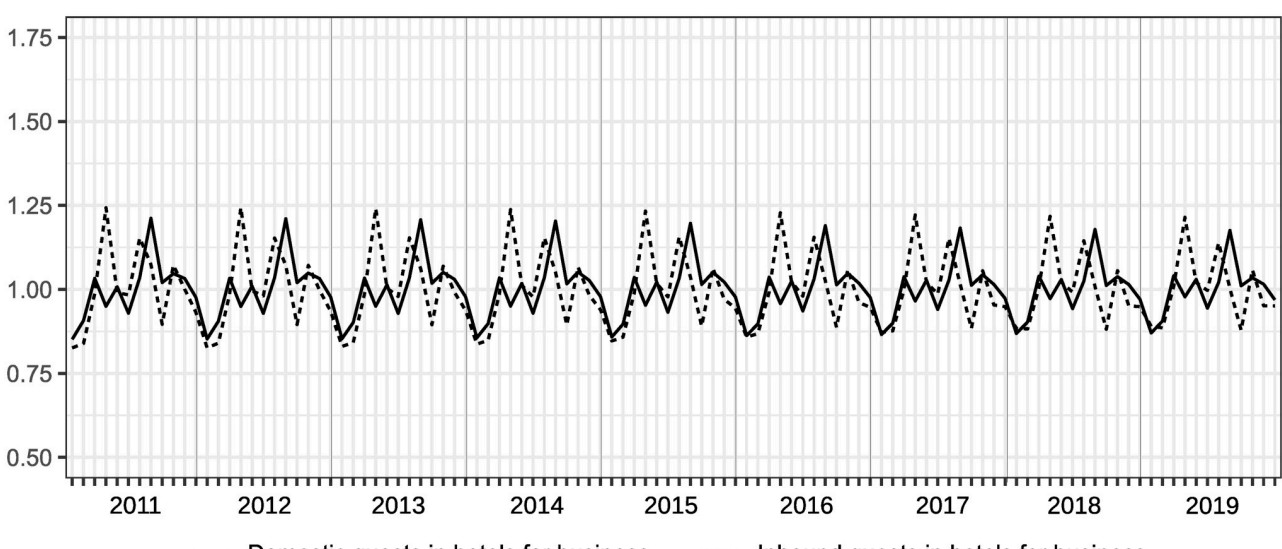

**Fig 4. Seasonal factors of overnight guests in hotels for business over 2011 to 2019.**

and inbound guests do not overlap. The seasonal factors of domestic guests for tourism in August declined from 2011 to 2019 as seen in Fig 3. We can also see changes in seasonal patterns in inbound guests for tourism, reflected by the increase in seasonal factors in February. In comparison with changes in seasonal factors for tourism, changes in seasonal factors for business are relatively small. However, observing the results of the F-test of moving seasonality, we can obtain different impressions from the figures. It is reported that the results of Freedman's test on the null of the absence of seasonality are that all items in Figs 3 and 4 are significant at 5%. The results of the F-test on the null of the absence of moving seasonality derived from X13-ARIMA-Seat with the X11 option are that domestic guests in hotels for business and inbound guests in hotels for tourism are significant at 5% and the F-values for those are 2.48 and 3.02, respectively (the critical value at the 5% is $F(8, 88) = 2.05$). Therefore, even though Figs 3 and 4 shows that there is moving seasonality for domestic guests in hotels for tourism, we can find that it is difficult to claim that from the viewpoint of the moving seasonality test. In addition, it is difficult to find clear moving seasonality for domestic guests in hotels for business, although the item is significant in the F-test. Furthermore, the seasonal pattern in inbound guests for tourism obviously changed from 2011 to 2019, and the movement was significant on the F-test.

Given the resemblance of seasonal patterns between overnight guests in hotels for tourism and business, this study shows results by prefecture obtained only from overnight stays in hotels for tourism.

The seasonal factors in Fig 3 are calculated from the total sum of overnight guests in hotels for tourism in Japan. However, there are time series data for overnight guests by 47 prefectures in Japan, and seasonal factors for each prefecture can be derived, which means that 47 interval-valued seasonal indices and graphs illustrating them can be generated. Figs 5 and 6 provide interval-valued seasonal indices of domestic and inbound guests for tourism in Hokkaido, Tokyo, and Okinawa as examples. Hokkaido is the northernmost prefecture in Japan, and Okinawa is the southernmost prefecture in Japan. Interval-valued seasonal indices are illustrated as error bars, and the mid-points of the ranges are connected with a line in the figures.

The ranges of the interval-valued seasonal index in Tokyo are smaller than those in Hokkaido and Okinawa as a whole, which is not surprising because Tokyo is the most developed city in Japan and has abundant tourism resources that are immune to seasonal fluctuations. Comparing intervals among months, combinations of months with wider ranges and months with narrower ranges differ by prefecture. Moreover, the ranges of intervals for inbound guests tend to be wider than those for domestic guests in these prefectures. In terms of the shape of the graph, the shapes of lines among the prefectures in Fig 5 are more similar than those in

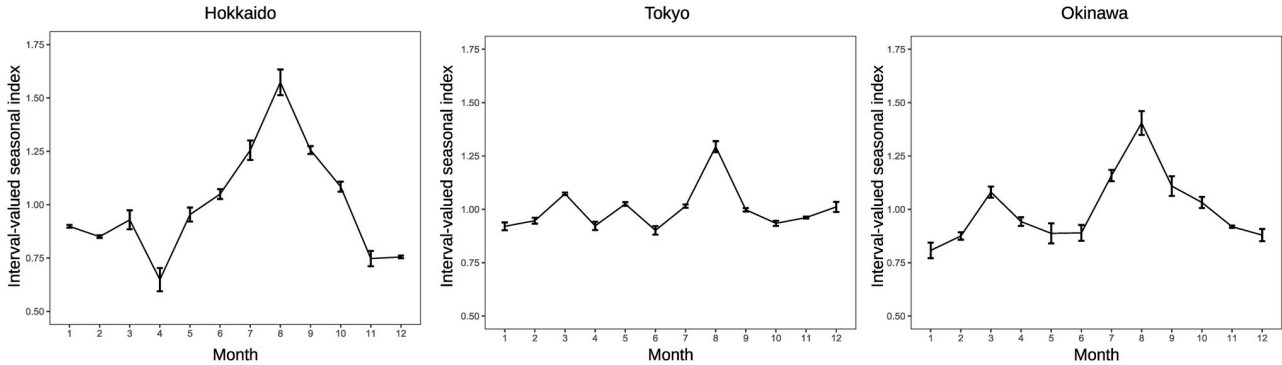

**Fig 5. Interval-valued seasonal indices of domestic guests for tourism in Hokkaido, Tokyo and Okinawa.**

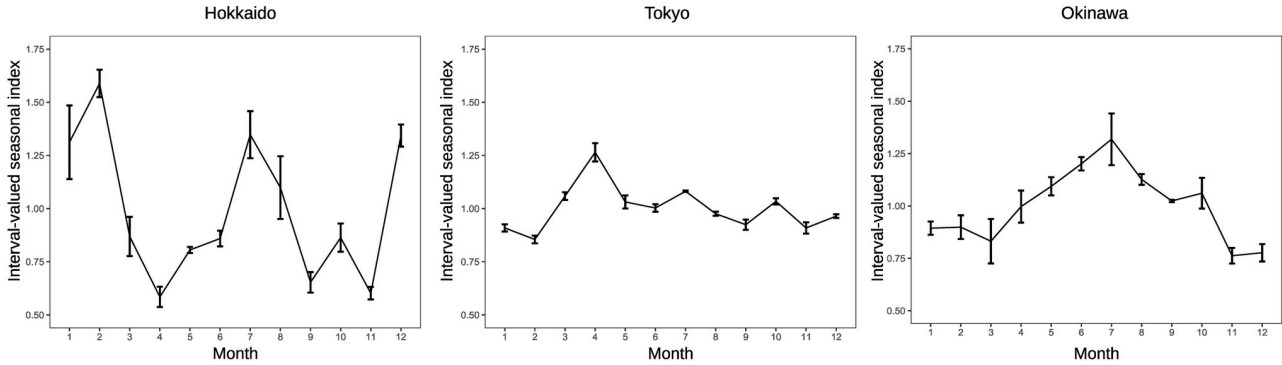

**Fig 6. Interval-valued seasonal indices of inbound guests for tourism in Hokkaido, Tokyo and Okinawa.**

Fig 6. This suggests that the diversity of seasonal patterns in inbound guests is stronger than that in domestic guests in Japan. We find that the impression is not wrong by viewing Figs 7 and 8, which illustrate ranges between maximum of seasonal factors and minimum of seasonal factors among prefectures on each month over 2011 to 2019 in the case of domestic guests and inbound guests for tourism.

This study also provides results of the F-test of moving seasonality by prefecture. Prefectures with significant F-values of 5% were Hokkaido (7.07), Aomori (3.36), Hukushima (2.50), Ibaraki (2.16), Tochigi (3.25), Gunma (2.38), Saitama (2.79), Chiba (2.67), Yamanashi (3.03), Nagano (2.15), Aichi (2.11), Hyogo (2.14), Nara (3.31), Wakayama (2.24), Shimane (2.18), Kagawa (2.51), Kumamoto (4.95), and Okinawa (2.66), 18 prefectures. With regard to inbound guests, the prefectures whose F-values are significant at 5% are Akita (2.74), Tochigi (2.75), Tokyo (3.08), Ishikawa (3.42), Fukui (2.84), Yamanashi (3.48), Gifu (5.10), Shiga (3.49), Osaka (2.38), Shimane (2.54), Hiroshima (2.25), Tokushima (3.13), Kochi (2.07), Saga (3.22), Nagasaki (4.51), Kumamoto (3.54), Miyazaki (3.37), and Okinawa (6.00), 18 prefectures. The values in parentheses correspond to the F-values. From these results, the seasonality of domestic guests in Hokkando and Okinawa shown in Fig 5 changes at the 5% significance level. Contrarily, the seasonality of inbound guests in Hokkando does not move at the 5% significance

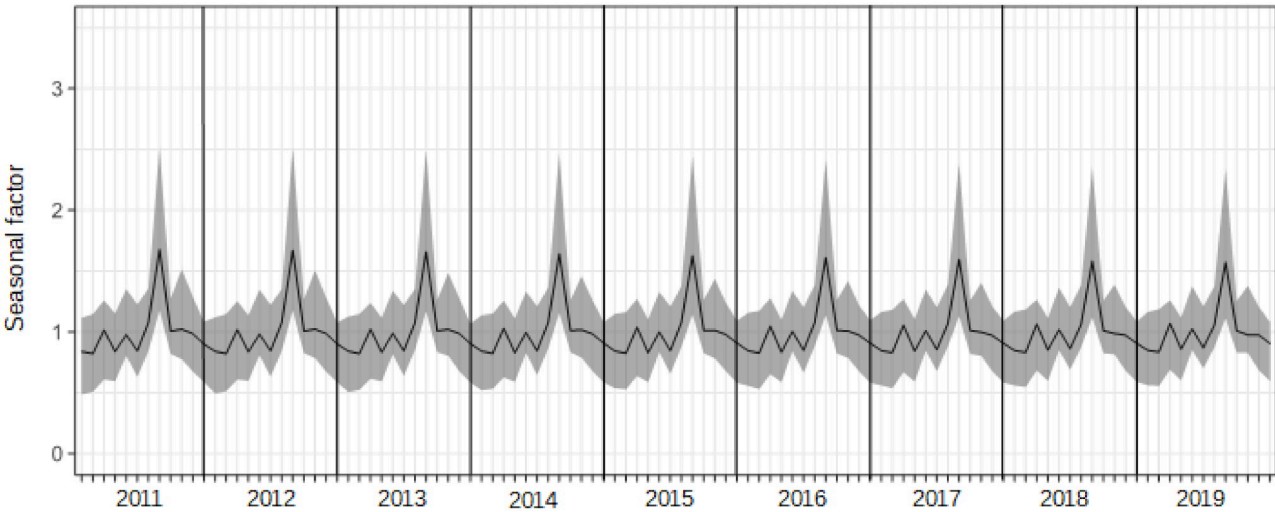

**Fig 7. Range between maximum and minimum of seasonal factors among prefectures in domestic guests for tourism.**

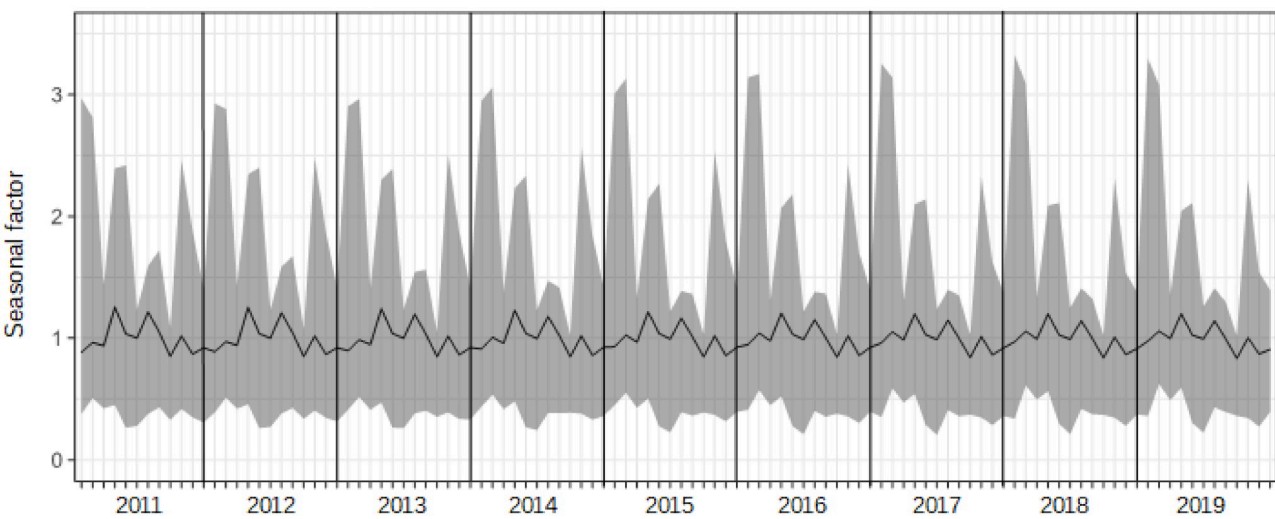

**Fig 8. Range between maximum and minimum of seasonal factors among prefectures in inbound guests for tourism.**

level, although its seasonal pattern seems not to be stable, as shown in Fig 6. The seasonality of inbound guests in Tokyo and Okinawa shown in Fig 6 significantly changes at the 5% level. An F-value does not necessarily reflect an impression obtained by the shape of the graph denoting the interval-valued seasonal index because the F-value is affected by not only the change in seasonal factors but also the change in residual components in Eq (1).

### Clustering patterns of Japanese seasonality

**Using interval-valued seasonal indices.** The clustering method mentioned in the previous section is applied to the interval-valued seasonal indices of 47 prefectures to classify seasonal patterns. The dendrograms obtained from interval-valued seasonal indices in domestic guests and inbound guests are shown in Figs 9 and 10, respectively. The seasonal patterns are divided into four clusters in both domestic and inbound guests. The four clusters for domestic guests are named D1, D2, D3, and D4, and the inbound guests are named I1, I2, I3, and I4. The cluster names are sequentially allotted to divided clusters from top to bottom in the dendrograms, as shown in Figs 7 and 8.

Traditional analysis of variance cannot be used to observe the characteristics of the clusters in this study because the applied data are interval-valued data, that is, non-traditional data. The interval-valued mean of each month is illustrated in a simple manner to see the characteristics. Figs 11 and 12 show the interval-valued mean of each month for the clusters.

Clusters D1 and D2 differ from D3 and D4 based on the strength of the main seasonal peak in August. Clusters D2 and D3 are different from D1 and D4 in the form of off-seasons, which are in January and February. In addition, the upper bounds of the intervals in May in D2 and D3 are slightly smaller than lower bounds in March in D2 and D3, and in D1 and D4. Further, the magnitude relationship between March and May is reversed. Hence, clusters D1, D2, D3, and D4 may be determined by the combination of the sizes of these interval-valued seasonal indices. However, all forms of the graphs for clusters D1, D2, D3, and D4 are close to each other, even considering the stability of the seasonality.

It is difficult to divide them into well-balanced clusters in terms of inbound guests, as shown in the dendrogram in Fig 10, because there are few prefectures with significantly diverging seasonal patterns from others. I1 and I2 are almost the same clusters, and the seasonal

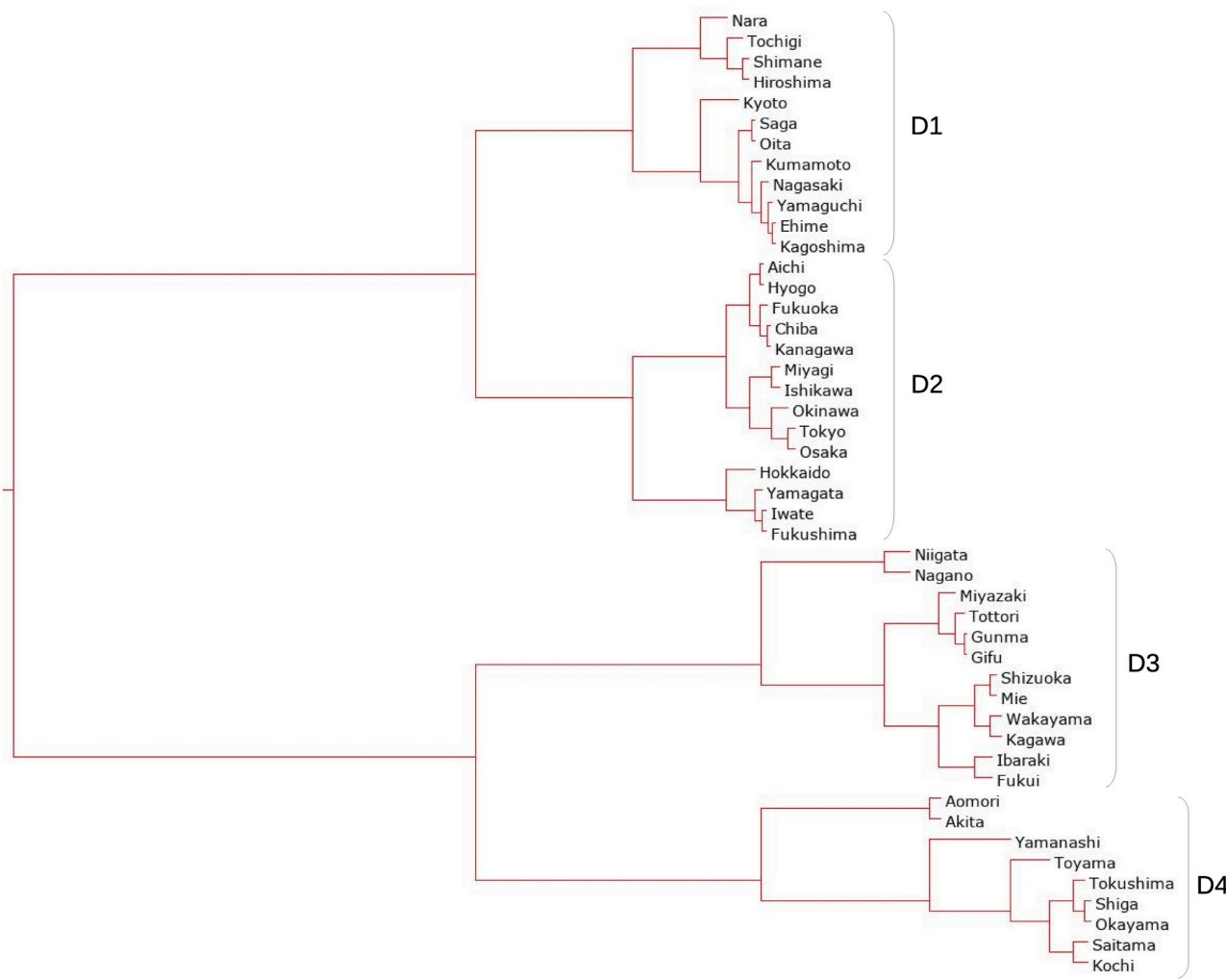

**Fig 9. A dendrogram for interval-valued seasonal indices in domestic guests for tourism.**

peaks occur in January and February. The peak of seasonality of I3 occurs in October, and the interval of the peak is relatively wide. Hence, the strength of the seasonal peak of I3 is not stable. We find that the seasonal peak of cluster I4 is in April; however, the seasonal scale of I4 is relatively weak.

Figs 13 and 14 show locations of the prefectures belonging to the clusters for domestic guests and inbound guests in a map of Japan, respectively. The figures are derived by a package, choroplethr (ver. 3.7.0) on R (ver. 4.0.3). The locations of prefectures belonging to same clusters for domestic guests are scattered across Japan and we cannot find obvious seasonal features from the map. Fig 14 presents clusters' positional features, I1 and I2, being almost the same clusters, are located near each other. Further, prefectures of I3 are agglomerated at a part of the north area of Japan.

**Using traditional seasonal indices.** We can also use another combination of seasonal indices and Ward's method to generate dendrograms and make clusters. As a comparative study, dendrograms obtained through the seasonal indices of domestic guests and inbound guests for Japanese prefectures and by the traditional Ward's method are shown in Figs 15 and 16, respectively. The clustering computation are conducted by the hclust function on R

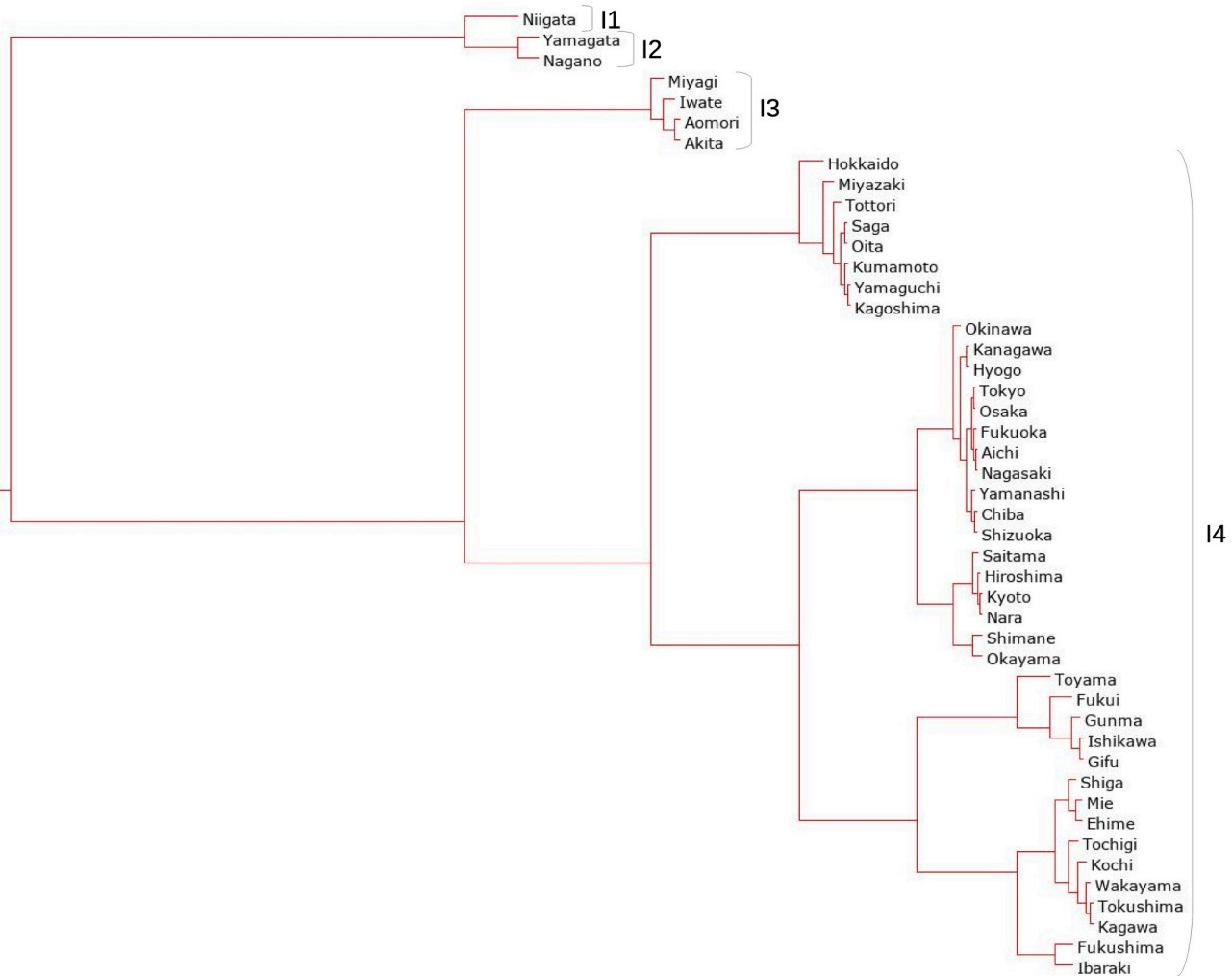

**Fig 10. A dendrogram for interval-valued seasonal indices in inbound guests for tourism.**

(version 4.0.3). Figs 17 and 18 show the mean of each month for the clusters. The seasonal patterns are divided into four clusters to better compare them with the clustering results obtained by interval-valued seasonal indices.

The clusters for domestic guests seem to be distinguished by the strength of seasonality in the main season (August) in Fig 17, similar to the clustering results obtained by interval-valued seasonal indices. Cluster WD3 displays a shoulder season in January and February, which is a different characteristic from the clustering results obtained through seasonal indices. Clusters WI1 and WI2 are the same as I1 and I2, respectively. However, we cannot find a cluster for inbound guests for whom the main peak season is October.

## Discussion

The seasonal peak of domestic guests is in August, which includes summer holidays, and graphical forms of interval-valued means for domestic guests by prefectures broadly resemble each other. In addition, the patterns were stable from 2011 to 2019. This indicates that domestic demand, occupying the majority of all Japanese tourism demand, is stably concentrated in limited periods, and this phenomenon occurs across Japan. From these results, it can be

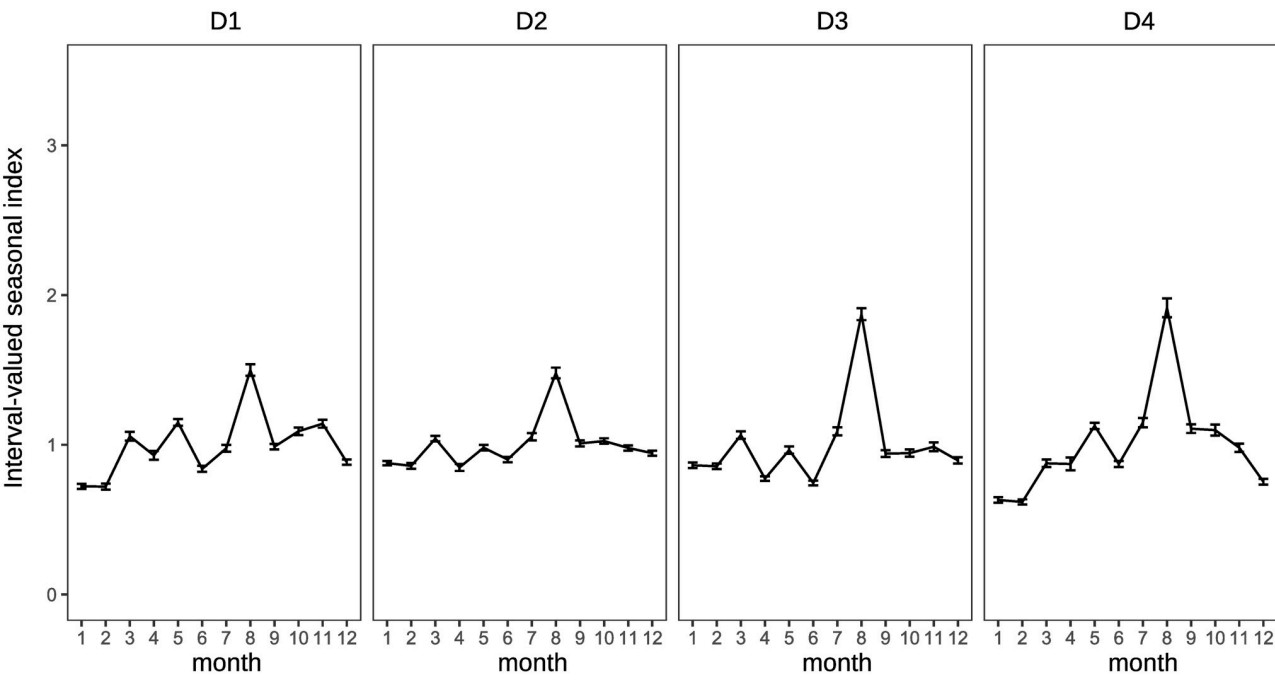

**Fig 11. Interval-valued mean of each cluster for domestic guests.**

considered that this problem might be caused by institutional factors that affect an entire nation's tourism demand. Hence, bold institute building for affecting institutional factors and customs, for example, staggering holidays might be necessary to more evenly distribute demand into other demand peaks throughout the year. The Japan Tourism Agency has already

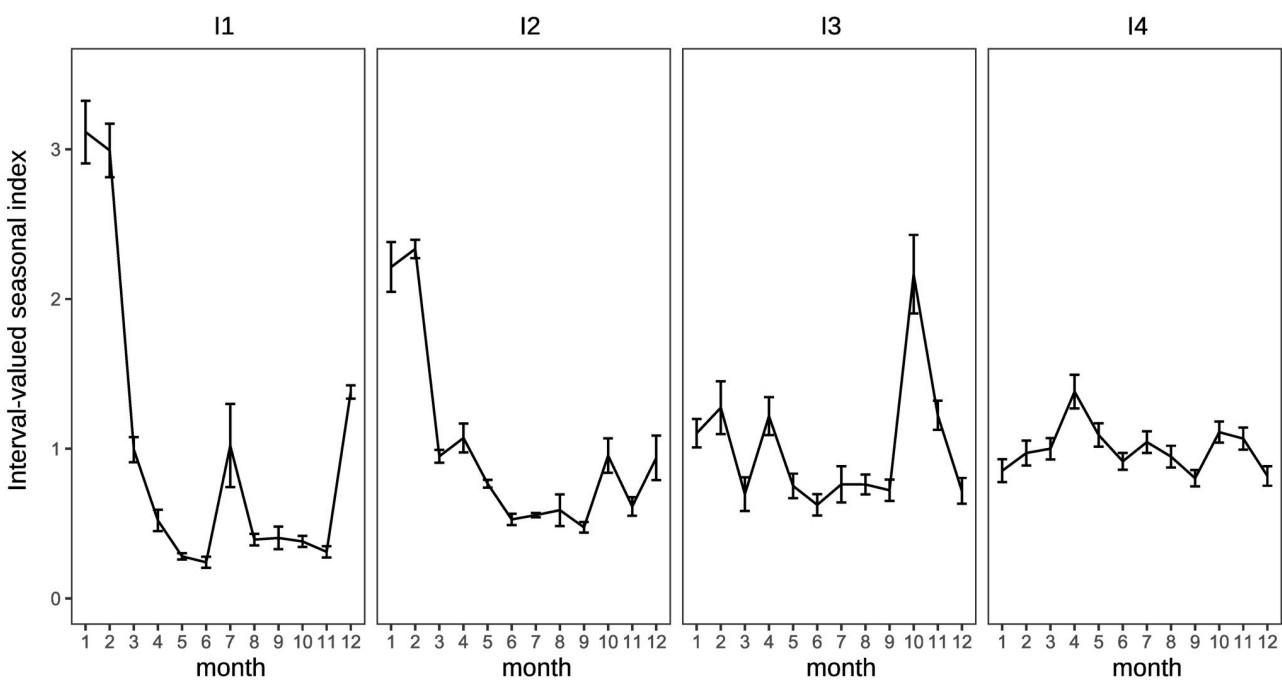

**Fig 12. Interval-valued mean of each cluster for inbound guests.**

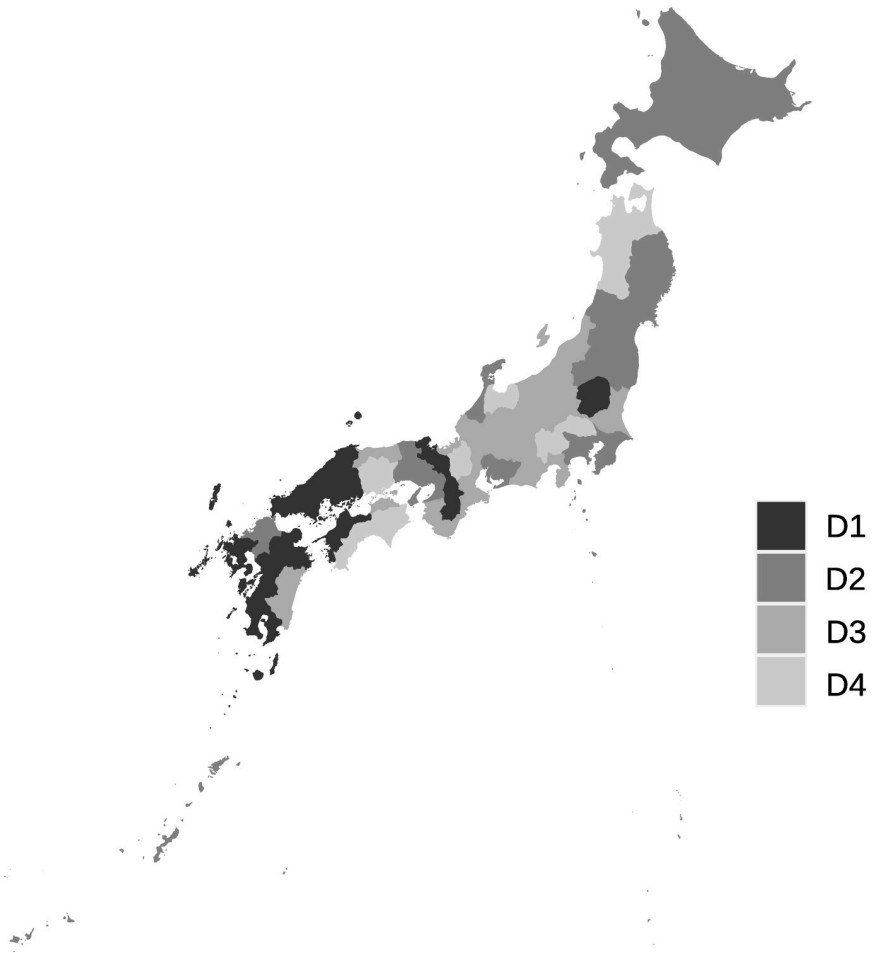

**Fig 13. Prefectures belonging to clusters for domestic guests.**

been tackled for staggering vacations and equalizing tourism demand since 2009 through a working team. In addition, the Japan Tourism Agency has implemented a project that aims to disperse tourism demand by time and place to avoid congestion in order to reduce the prevalence of COVID-19 in 2020 [38]. After fully containing the COVID-19 pandemic, it is possible that this project could be useful in reducing the concentration of domestic tourism demand.

Prefectures with similar seasonal patterns of inbound guests are adjacent to each other in a map of Japan. Prefectures belonging to clusters I1 and I2, whose peak is in the snow season of January and February, have famous ski resorts that are frequented by inbound tourists (see Fig 1, p.138 in [39]). Hokkaido prefecture not belonging to I1 and I2 also has famous ski resorts areas, such as the Niseko-Hirafu district [40]. Indeed, Fig 6 shows that there is a seasonal peak of inbound guests in January and February in Hokkaido from 2011 to 2019. Hokkaido is one of the most famous prefectures for tourism in Japan and includes large cities and famous tourism destinations other than ski resorts, such as the Shiretoko peninsula, registered as a UNESCO World Heritage Site. These results might lead to exclusion from clusters I1 and I2. A part of the north of Japan, which is a part of the Tohoku area, belongs to cluster I3. The Tohoku area was devastated by a magnitude 9.0 earthquake and a tsunami that followed it in March 2011. This disaster caused inbound guests to depart from the Tohoku area. Based on these circumstances, the Japan National Tourism Organization in Japan Tourism Agency

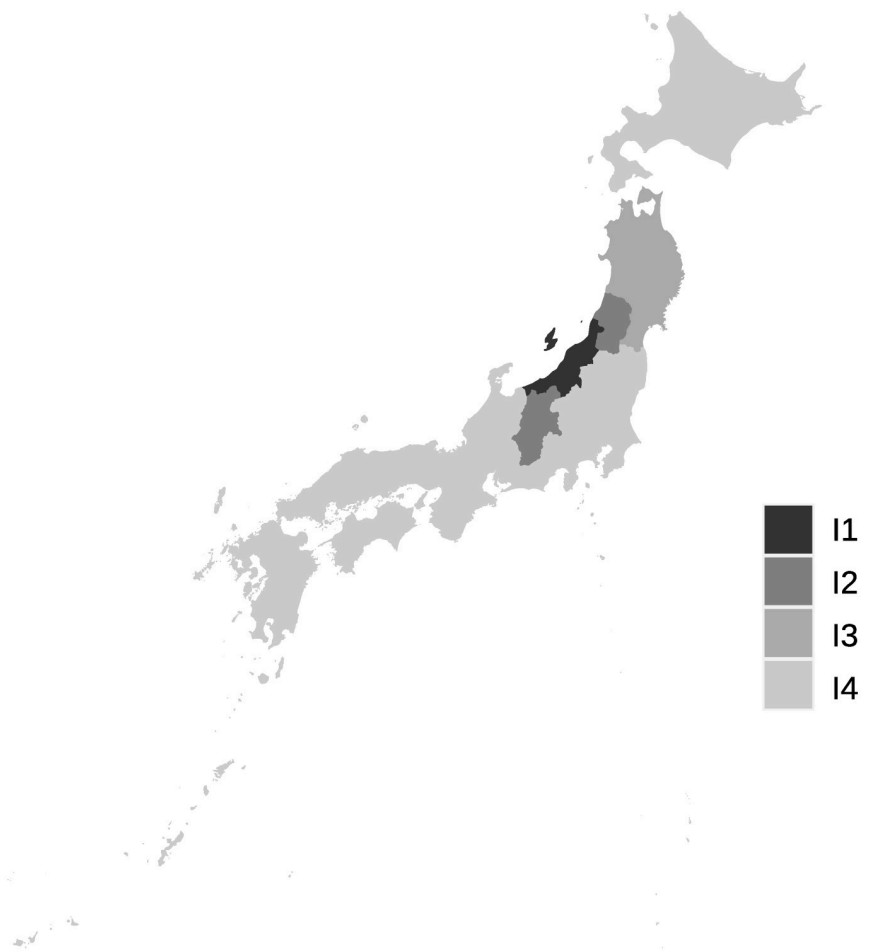

**Fig 14. Prefectures belonging to clusters for inbound guests.**

started the Tohoku destination campaign in 2016, which increased the inbound demand in the Tohoku region for earthquake restoration support. This campaign was the first worldwide destination campaign in Japan [41]. Important targeted markets for the campaign are mainly China, Taiwan, Thailand, and Korea in 2016 and 2017 [42, 43]. Consequently, the campaign's goal of achieving 1,500,000 inbound overnight guests in the Tohoku area by 2020 was attained in 2019. In Aomori, Iwate, Miyagi, and Akita prefectures belonging to cluster I3, the number of Chinese tourists was especially high. Specifically, the percentage of Chinese overnight guests in these four prefectures were 65.7%, 80.7%, 60.9%, and 62.3%, respectively (in 2019). Thus, a peak of I3 in October may stem from a long holiday in China, the National Day of the People's Republic of China. While modifying seasonal patterns of domestic demand seems to be difficult in Japan, it might be possible that some neighboring prefectures execute a marketing strategy for inbound tourism demand to change their seasonal patterns and build a common characteristic seasonal pattern. If the concentration of inbound tourism demand in the area needs to be reduced, then it might be effective to control tourism demand by each origin country through a marketing promotion in the whole area. This implies that seasonal patterns of inbound demand in Japan may change if the demand structure of the origin countries is reconstructed after the COVID-19 pandemic.

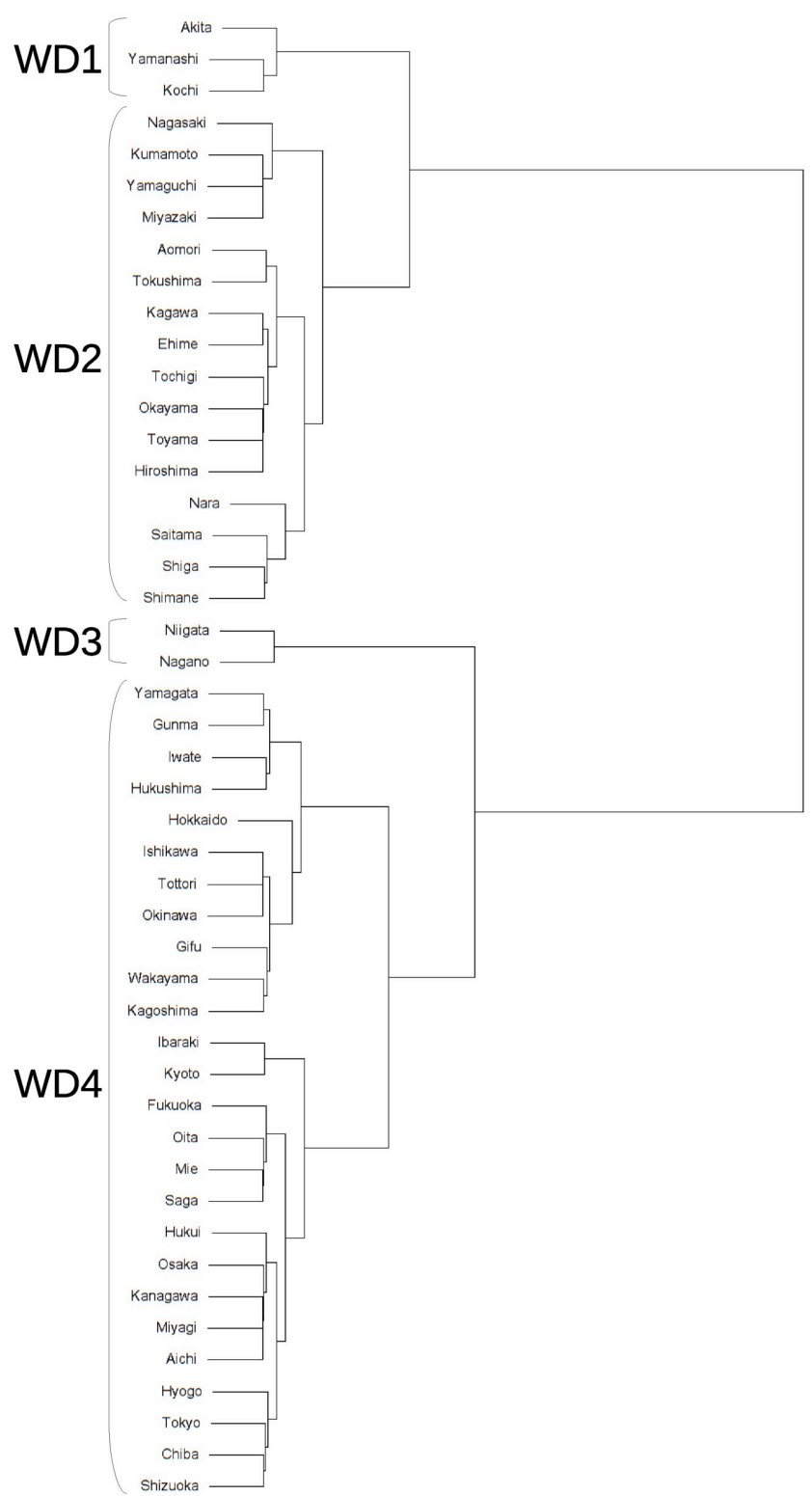

**Fig 15. A dendrogram for seasonal indices in domestic guests for tourism.**

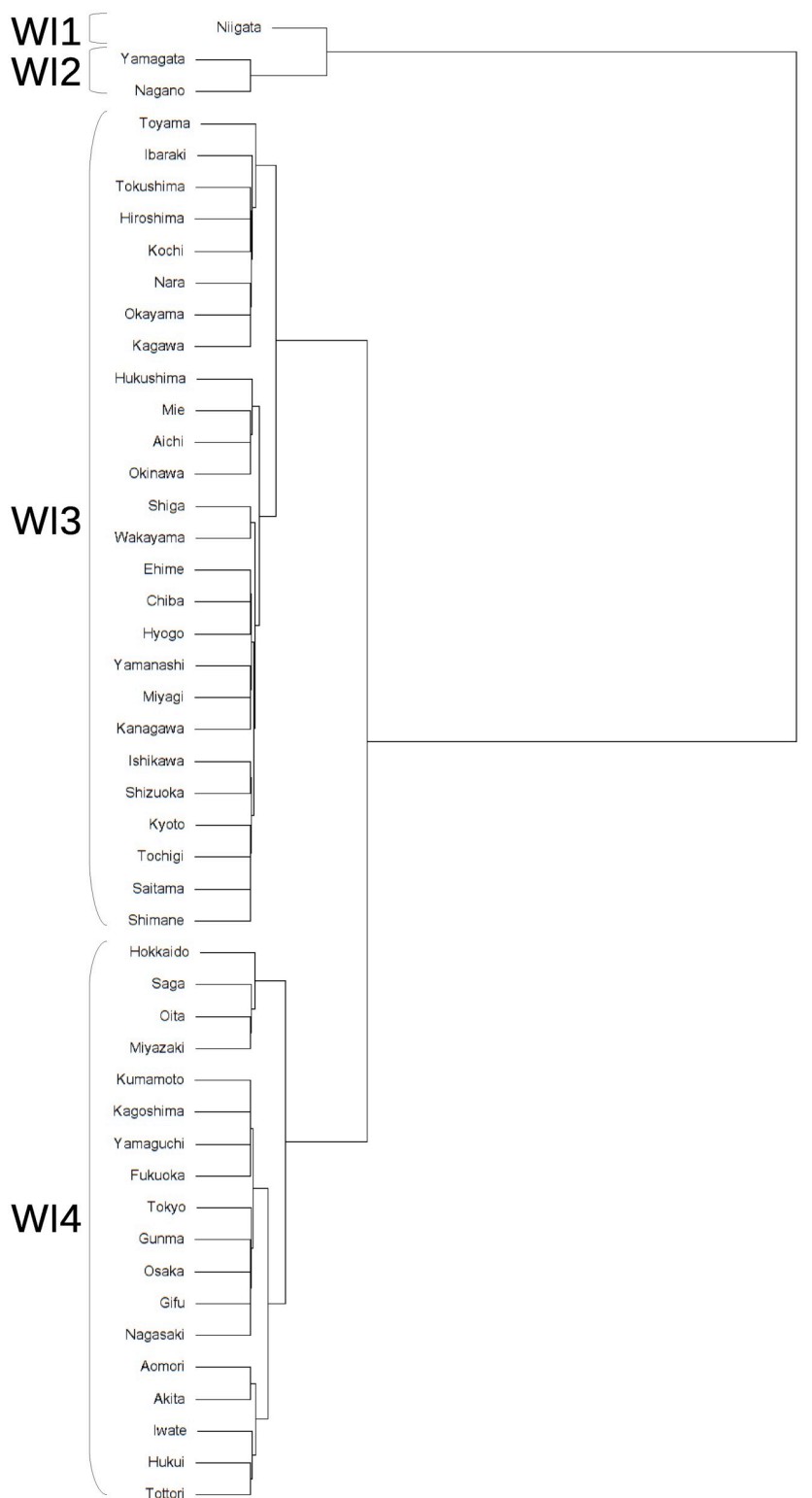

**Fig 16. A dendrogram for seasonal indices in inbound guests for tourism.**

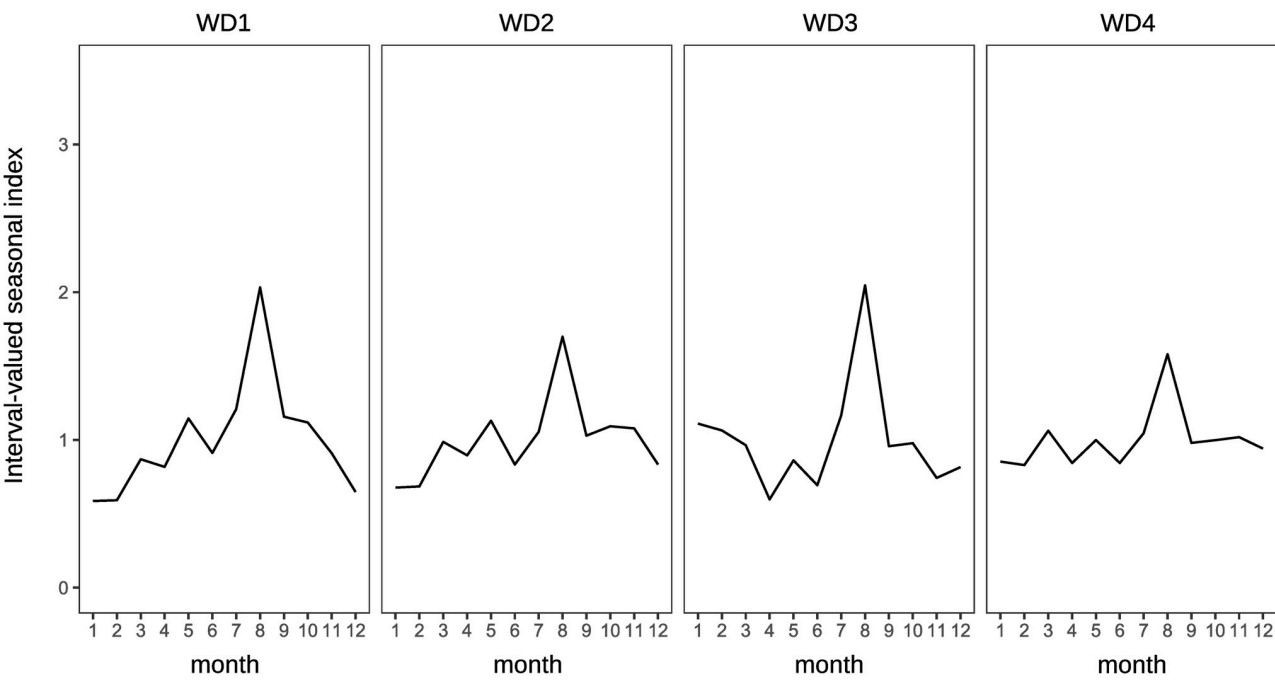

**Fig 17. Mean of each cluster for domestic guests.**

We find that characteristics of several clusters obtained by traditional seasonal indices and Ward's method differ from ones derived from interval-valued seasonal indices. Cluster WD3, which consisted of Niigata and Nagano, shows that these prefectures have a small peak in winter season as a slight seasonal feature. These prefectures are famous across Japan for their

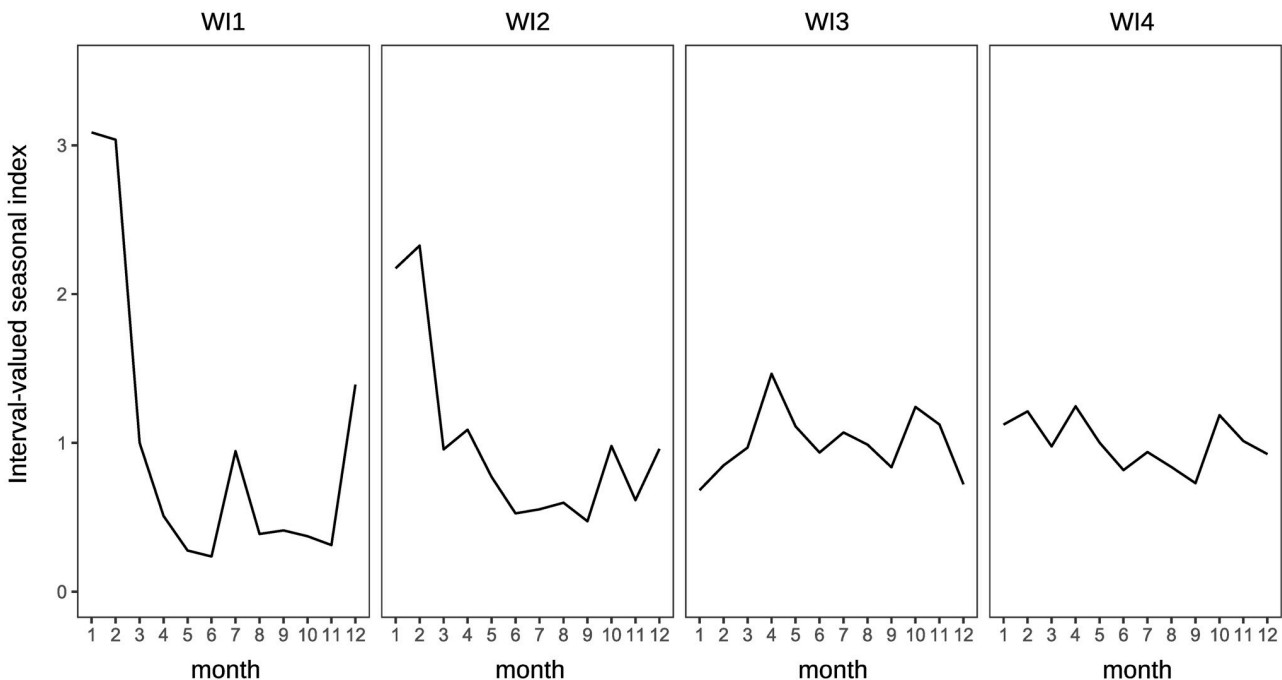

**Fig 18. Mean of each cluster for inbound guests.**

many ski resorts. However, there is no major difference in domestic demand. This may be due to the insignificance of seasonal changes in domestic demand. In terms of inbound demand, the two clusters: WI1 and WI2 which are the same as I1 and I2 are generated. This means that these prefectures have a robust characteristic seasonality for inbound demand. In addition, it is worth noting that a cluster which includes prefectures in the Tohoku area and has a main peak in October cannot be observed using seasonal indices. This may be because the increase in inbound demand in the Tohoku region is rapid, as already mentioned. Although this rapid change cannot be captured by a seasonal index, an interval-valued seasonal index can capture it, as seen in Fig 12.

## Conclusions

This study proposes a new seasonal measurement, namely the interval-valued seasonal index, which is based on interval-valued data in symbolic data. Additionally, a hierarchical clustering method for interval-valued data based on Ward's method, which was suggested by Ogasawara and Kon [15], is introduced to grasp the whole picture of a dataset comprising many interval-valued seasonal indices and classify their seasonal patterns. By using the seasonal measurement and clustering method, the computational results obtained from Japanese overnight guest data from 2011 to 2019 are shown as a case study. These results demonstrate that there are differences in not only seasonal patterns but also their stability between domestic and inbound demand. In domestic tourism demand, seasonal patterns are almost the same among prefectures and their stability is stiff even though more than one-third of prefectures have significant F-values on the seasonality moving test at the 5% level. In contrast, seasonal patterns of inbound tourism demand among prefectures are more diverse than those of domestic tourism demand. We find characteristic seasonal patterns as clusters consisting of adjusted prefectures with ski resort developed regions and the Tohoku area in Japan. In particular, the cluster of the Tohoku area having rapidly recovered from the earthquake can be obtained with the proposed new measurement, the interval-valued seasonal index, and not with the traditional seasonal index. A tendency of inbound overnight guests moving more easily than domestic overnight guests, especially during peak months, can be found in Italy from 1988 to 2004 [17]. When we deal with the seasonality of inbound demand with low stability, an interval-valued seasonal index might be a useful seasonal measurement. Furthermore, there are data-analysis methods for interval-valued data other than the hierarchical clustering method introduced in this study [14, 28, 44]. Thus, the interval-valued seasonal index can be utilized to analyze seasonal patterns by various other methods.

In terms of limitations of this study, an interval-valued seasonal index does not have transition information, specifically going up or down for each month. The results of the Japanese case study cannot include this information. Furthermore, an interval-valued seasonal index corresponds to a prefecture that covers a large area, several markets, and various tourism resources, which limits the interpretation of analysis results. If an analysis subject corresponding to an interval-valued seasonal index becomes smaller and finer, then the procedure presented in this study is expected to provide more specific insights for tourism policy decisions or building marketing strategies. However, the computational cost of conducting the cluster analysis suggested in this study greatly increases due to the large number of given interval-valued data, as the clustering method does not use the squared Euclidean distance. Regardless of the numbers shown in the Japanese case study, we need to choose a moderate sample size to apply the procedure introduced in this study.

While this study focuses on seasonal patterns and expresses their stability by interval-valued data, tourism studies have also focused on amplitude of seasonality, using seasonal range and

ratio. From the viewpoint of symbolic data analysis, the seasonal range is equivalent to the width of an interval-valued data whose lower and upper boundaries correspond to the minimums and maximums of the seasonal factors in a year for an analysis subject. Therefore, we can define further seasonal measurements to analyze tourism seasonality. As a future study, we could consider reviewing contribution factors for seasonality, although we might need to develop further statistical models [44, 45].

## Acknowledgments

I appreciate the helpful comments from reviewers. They greatly improved this manuscript.

## Author Contributions

**Conceptualization:** Yu Ogasawara.

**Formal analysis:** Yu Ogasawara.

**Funding acquisition:** Yu Ogasawara.

**Investigation:** Yu Ogasawara.

**Methodology:** Yu Ogasawara.

**Project administration:** Yu Ogasawara.

**Resources:** Yu Ogasawara.

**Software:** Yu Ogasawara.

**Supervision:** Yu Ogasawara.

**Validation:** Yu Ogasawara.

**Visualization:** Yu Ogasawara.

**Writing – original draft:** Yu Ogasawara.

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
