## [Decision Letter · Decision Letter 0]

15 Feb 2022

PONE-D-21-39279New seasonal measurement with stability and clustering seasonal patterns: a case study in Japan from 2011 to 2019PLOS ONE

Dear Dr. Ogasawara,

Thank you for submitting your manuscript to PLOS ONE. After careful consideration, we feel that it has merit but does not fully meet PLOS ONE’s publication criteria as it currently stands. Therefore, we invite you to submit a revised version of the manuscript that addresses the points raised during the review process.

We look forward to receiving your revised manuscript.

Kind regards,

Sathishkumar V E

Academic Editor

PLOS ONE

Journal Requirements:

(This work was supported by JSPS KAKENHI Grant Number JP20K20080.)

4. We note that Figures 12 and 13 in your submission contain map images which may be copyrighted. All PLOS content is published under the Creative Commons Attribution License (CC BY 4.0), which means that the manuscript, images, and Supporting Information files will be freely available online, and any third party is permitted to access, download, copy, distribute, and use these materials in any way, even commercially, with proper attribution. For these reasons, we cannot publish previously copyrighted maps or satellite images created using proprietary data, such as Google software (Google Maps, Street View, and Earth). For more information, see our copyright guidelines: http://journals.plos.org/plosone/s/licenses-and-copyright.

a. You may seek permission from the original copyright holder of Figures 12 and 13 to publish the content specifically under the CC BY 4.0 license.  

Reviewers' comments:

Reviewer's Responses to Questions

**Comments to the Author**

1. Is the manuscript technically sound, and do the data support the conclusions?

Reviewer #1: Partly

Reviewer #2: Yes

2. Has the statistical analysis been performed appropriately and rigorously? 

Reviewer #1: Yes

Reviewer #2: Yes

3. Have the authors made all data underlying the findings in their manuscript fully available?

Reviewer #1: No

Reviewer #2: Yes

4. Is the manuscript presented in an intelligible fashion and written in standard English?

Reviewer #1: No

Reviewer #2: Yes

5. Review Comments to the Author

Reviewer #1: This paper study the seasonal patterns of Japanese overnight data from 2011 to 2019. The measurement is based on a seasonal index and expressed interval-valued data, which are a kind of symbolic data. This paper is a study paper and the contributions are minimal, whereas the work and results are promising and fit for our journal. The organization of the paper is well. However, there are certain corrections required before consider this paper for publication.

1. The literature of this article is poor. It is recommended to provide the literature on the topic. The motivation behind the topic also needed to include.

2. It is recommended to summarize the contributions in the introduction.

3. Section 3 is too small, it is recommended to elaborate through detailed discussion and importance of each equation.

4. There are several performance metrics in the literature, whereas the authors considered only few. It is recommended to evaluate through multiple metrics.

5. It is recommended to provide the futuristic prediction about the seasons.

6. What are the limitations identified during your study?

Reviewer #2: 1. Need to add a Comparison study.

2. Need to update the limitation.

3. How to deploy the research idea, detailed information required.

4. In many figures the X-Axis name was inverted, kindly reupdate it.

5. Overall the idea was good.

6. The dataset is available in public, easily we can able to downloadable. That's great work.

6. PLOS authors have the option to publish the peer review history of their article (what does this mean?). If published, this will include your full peer review and any attached files.

Reviewer #1: No

Reviewer #2: **Yes: **ANANDHAN K

---

## [Author Response · Author response to Decision Letter 0]

31 Mar 2022

Dear Dr. Sathishkumar V E

Thank you for inviting me to submit a revised manuscript. I appreciate the time and effort you and each of the reviewers have dedicated to providing insightful feedback to my manuscript. The comments have helped significantly improve the manuscript.

The following is a point-by-point response to the reviewers' comments. 

Response to Reviewer #1:

I wish to express my appreciation to the reviewer for insightful comments on our manuscript. The comments have helped significantly improve the paper.

The following is a point-by-point response to the reviewers' comments. 

1. The literature of this article is poor. It is recommended to provide the literature on the topic. The motivation behind the topic also needed to include.

Thank you for your thoughtful suggestion. I added some references on this topic to express more details and clarify the motivation. Specifically, I added sentences on lines 76-85, 144-147, 151-159 and 170. 

2. It is recommended to summarize the contributions in the introduction.

I agree with your suggestion. I added a paragraph to mention the contribution of this study on lines 86-94. This contribution includes a result of the comparison study which is newly added in the revised manuscript. 

3. Section 3 is too small, it is recommended to elaborate through detailed discussion and importance of each equation.

Thank you for your suggestion. I explained Eq. (3) on lines 182-184 and added sentences and a figure on lines 192-201 to discuss for details. 

4. There are several performance metrics in the literature, whereas the authors considered only few. It is recommended to evaluate through multiple metrics.

Since the individual data (prefectures) in the Japanese case study does not have correct answer labels, it is difficult to derive performance metrics for an evaluation. Hence, I additionally conducted a clustering analysis with traditional seasonal index and Ward’s method to compare with the results of a suggested method and evaluate by obtained insights. I explained these results in the section named “Using traditional seasonal indices” in page 16 and on lines 411-416 and 426-439. 

5. It is recommended to provide the futuristic prediction about the seasons.

I appreciate your comment on this point. Accordingly, I have added sentences on lines 390-392 and lines 423-425. I have mentioned the futuristic prediction of Japanese seasonality, based on the results of this study. 

6. What are the limitations identified during your study?

I thank the reviewer for this comment. I additionally mentioned the limitations on lines 466-468 and 474-478. 

Response to Reviewer #2:

I wish to thank the reviewer for thoughtful comments on my manuscript. The comments have helped significantly improve the paper. The following is a point-by-point response to the reviewers' comments.

1. Need to add a Comparison study.

I wish to thank for this comment. The results of the additional work are shown in the section named “Using traditional seasonal indices” with Fig 15, 16, 17 and 18 in page 16. The discussion for the results were mentioned on lines 411-416 and 426-439.

2. Need to update the limitation.

Thank you for this comment. I additionally mentioned the limitations on lines 466-468 and 474-478.

3. How to deploy the research idea, detailed information required.

I appreciate the comment on this point. I added a future issue and its details on lines 479- 487. 

4. In many figures the X-Axis name was inverted, kindly reupdate it.

Thank you for the review’s suggestion. I confirmed that many figures were automatically rotated when making a combined pdf file in the submitting process. Specifically, Fig. 1, 2, 3, 4, 5, 6, 7, 8, 9, 10, 11, 12, 17 and 18 were rotated at 90 degrees in the resubmitted pdf file. I consider that this problem might be caused by the submitting system.

5. Overall the idea was good.

6. The dataset is available in public, easily we can able to downloadable. That's great work.

Thank you so much for these comments. As the review’s pointing out, all results in this study are derived from Japanese government open data. I am glad to hear that.

Sincerely,

Yu Ogasawara

---

## [Decision Letter · Decision Letter 1]

11 Apr 2022

New seasonal measurement with stability and clustering seasonal patterns: a case study in Japan from 2011 to 2019

PONE-D-21-39279R1

Dear Dr. Ogasawara,

We’re pleased to inform you that your manuscript has been judged scientifically suitable for publication and will be formally accepted for publication once it meets all outstanding technical requirements.

Kind regards,

Sathishkumar V E

Academic Editor

PLOS ONE

Additional Editor Comments (optional):

Reviewers' comments:

Reviewer's Responses to Questions

**Comments to the Author**

1. If the authors have adequately addressed your comments raised in a previous round of review and you feel that this manuscript is now acceptable for publication, you may indicate that here to bypass the “Comments to the Author” section, enter your conflict of interest statement in the “Confidential to Editor” section, and submit your "Accept" recommendation.

Reviewer #1: All comments have been addressed

Reviewer #2: All comments have been addressed

2. Is the manuscript technically sound, and do the data support the conclusions?

Reviewer #1: Yes

Reviewer #2: Yes

3. Has the statistical analysis been performed appropriately and rigorously? 

Reviewer #1: Yes

Reviewer #2: Yes

4. Have the authors made all data underlying the findings in their manuscript fully available?

Reviewer #1: No

Reviewer #2: Yes

5. Is the manuscript presented in an intelligible fashion and written in standard English?

Reviewer #1: Yes

Reviewer #2: Yes

6. Review Comments to the Author

Reviewer #1: The authors addressed all the recommended comments and this version may be considered for publication in this journal.

Reviewer #2: The author addressed all the reviewer comments. The manuscript is written well in standard English.

All the graphs are plotted now in a proper manner.

7. PLOS authors have the option to publish the peer review history of their article (what does this mean?). If published, this will include your full peer review and any attached files.

Reviewer #1: No

Reviewer #2: **Yes: **ANANDHAN K

---

## [Editor Report · Acceptance letter]

13 Apr 2022

PONE-D-21-39279R1 

New seasonal measurement with stability and clustering seasonal patterns: a case study in Japan from 2011 to 2019 

Dear Dr. Ogasawara:

I'm pleased to inform you that your manuscript has been deemed suitable for publication in PLOS ONE. Congratulations! Your manuscript is now with our production department. 

Kind regards, 

on behalf of

Dr. Sathishkumar V E 

Academic Editor

PLOS ONE